# Synergistic Antihyperglycemic and Antihyperlipidemic Effect of Polyherbal and Allopolyherbal Formulation

**DOI:** 10.3390/ph16101368

**Published:** 2023-09-27

**Authors:** Yahya Alhamhoom, Syed Sagheer Ahmed, Rupesh Kumar M., MD. Salahuddin, Bharathi D. R., Mohammed Muqtader Ahmed, Syeda Ayesha Farhana, Mohamed Rahamathulla

**Affiliations:** 1Department of Pharmaceutics, College of Pharmacy, King Khalid University, Al Faraa, Abha 62223, Saudi Arabia; ysalhamhoom@kku.edu.sa; 2Department of Pharmacology, Sri Adichunchanagiri College of Pharmacy, Adichunchanagiri University, BG Nagara, Mandya 571448, India; rambha.eesh@gmail.com; 3Department of Pharmacology, Alameen College of Pharmacy, Bengalore 560027, India; manirupeshkumar@yahoo.in; 4Department of Chemistry, Alameen College of Pharmacy, Bengalore 560027, India; rcsalahuddin@alameenpharmacy.edu; 5Department of Pharmaceutics, College of Pharmacy, Prince Sattam Bin Abdul Aziz University, Al Kharj 11942, Saudi Arabia; muqtadernano@psau.edu.sa; 6Department of Pharmaceutics, Unaizah College of Pharmacy, Qassim University, Unaizah 51911, Saudi Arabia; a.farhana@qu.edu.sa

**Keywords:** allopolyherbal, diabetes, polyherbal, streptozotocin, rat, in vivo

## Abstract

Polyherbal formulation (PHF) enhances therapeutic efficacy and minimizes side effects by reducing individual herb dosages. Allopolyherbal formulation (APHF) combines polyherbal extracts with allopathic medication, effectively reducing the latter’s required dose and mitigating associated adverse effects. The current study intends to assess the anti-diabetic effects of PHF and APHF *in-vivo*. Dried raw powders of *Cassia auriculata* leaf, *Centella asiatica* leaf, and *Zingiber officinale* rhizome were extracted by cold maceration process using 70% ethanol. These extracts were combined in three different ratios to make PHF. PHF was subjected to qualitative and quantitative phytochemical investigations. APHF has been prepared by combining a potent ratio of PHF with metformin in three different ratios. The compatibility of APHF has been confirmed by differential scanning calorimetry (DSC). In vivo activity was also evaluated in streptozotocin-induced diabetic albino rats. PHF (3 different ratios at a dose of 200–400 mg/kg b.w), APHF (combination of PHF and metformin in 3 different ratios, 200 + 22.5, 200 + 45, and 200 + 67.5 mg/kg b.w), and metformin (90 mg/kg b.w) were administered to albino rats for 21 consecutive days. Blood glucose levels were estimated on the 1st, 7th, 14th, and 21st days of treatment. On the 21st day, blood was collected by cardiac puncture for biochemical analysis. The liver and pancreas were isolated and subjected to histopathological analysis. PHF and APHF showed significant anti-diabetic and antihyperlipidemic efficacy. In comparison to PHF, APHF had the most promising action. The current study demonstrated that PHF and APHF are safe and efficacious drugs in the treatment of diabetes mellitus as they help to replace or lower the dose of metformin, thereby decreasing the risks of metformin.

## 1. Introduction

Diabetes mellitus (DM) is a metabolic disorder characterized by disturbances in glucose, protein, and lipid metabolism. It is caused by either a lack of insulin secretion (T1DM) or a decreased sensitivity of the tissues to insulin (T2DM). In 2021, the worldwide prevalence of diabetes among individuals aged 20 to 79 stood at 537 million, with projections indicating an anticipated increase to 783 million by the year 2045 [1]. Diabetes canbe managed with exercise, dietary restrictions, insulin injections, and the use of oral hypoglycemic medicines [2].

Herbal remedies have risen in popularity in recent years due to their safety, efficacy, and affordability when compared to synthetic therapies. Polyherbal formulations (PHF) are made up of two or more herbs. The concept of polyherbalism appeared in Ayurvedic literature (Sharangdhar Samhita) in1300 AD. A polyherbal composition boosts medicinal efficacy while limiting side effects by lowering the concentration of individual herbs in a mixture [3]. Allopolyherbal formulation (APHF) is a polyherbal formulation combined with an allopathic drug [4]. The major goal of combining allopathic drugs and herbs is to reduce the adverse effects of allopathic drugs. Most allopathic drugs have one or more side effects. As a consequence, combining polyherbs with allopathic therapy removes or reduces the unfavorable effects of allopathic medications. Allopolyherbal formulations are much more efficient and effective in the case of chronic or long-term diseases or issues [5].

Metformin is a member of the biguanideclass of oral hypoglycemic agents that operates by reducing hepatic gluconeogenesis and increasing tissue insulin sensitivity. Metformin side effects include nausea, vomiting, metallic taste, diarrhea, anorexia, and a variety of other gastrointestinal issues. Biguanides may increase the risk of lactic acidosis in those who have a history of alcoholism, kidney disease, liver disease, cardiovascular disease, chronic cardiopulmonary disease, pregnancy, and breastfeeding. By reducing the dose of metformin, such side effects can be reduced or avoided. However, lowering the dose of metformin is still a difficult approach, but it may help to mitigate the unfavorable effects as much as possible [6]. Incorporating a minimal dose of metformin into allopolyherbal formulation (APHF) results in a remarkably potent reduction in blood glucose levels, equivalent to the impact achieved by higher doses of metformin when administered on its own. The primary advantages of this research proposal lie in the potential to mitigate unwanted side effects and reduce the required dosage of allopathic medication, particularly metformin, by substituting it with PHF or APHF.

*Cassia auriculata* L. (CAr), often known as “Tanner’s Cassia” (Ceasalpinaceae), is a medicinal plant found across India [7]. CAr leaves have long been used to treat ulcers, liver problems, jaundice, diabetes, and leprosy. It also possesses antispasmodic, antiviral, and antidiarrheal properties [8,9]. *Centella asiatica* (CAs) is a common Indian herb (Umbelliferae family). It has historically been used to treat a wide range of CNS disorders, such as schizophrenia, cognitive impairment, and epilepsy. It is also used to treat asthma, leprosy, jaundice, hepatitis, diarrhea, syphilis, and smallpox [10,11]. Ginger, or *Zingiber officinale* (ZO), is an underground rhizome of the Zingiberaceae family [12]. It is a nutritional supplement that is used to treat a wide range of diseases, including asthma, cough, indigestion, anorexia, and constipation [13].

*Cassia auriculata*, *Centella asiatica*, and *Zingiber officinale* have all been reported to lower blood glucose levels [14,15,16]. Despite the fact that each of these plants has potent anti-diabetic potential on its own, there is no conclusive evidence that the polyherbal formulation (a combination of *Cassia auriculata*, *Centella asiatica*, and *Zingiber officinale*) and the allopolyherbal formulation (a combination of polyherbal formulation and metformin) have a synergistic anti-diabetic effect.

Hence, the goal of the current investigation is to evaluate the antihyperglycemic and antihyperlipidemic effects of polyherbal and allopolyherbal formulations in streptozotocin-induced diabetic rats.

## 2. Results and Discussions

### 2.1. Phytochemical Analysis of Polyherbal Formulation (PHF)

Phytochemical analysis showed the presence of carbohydrates, proteins, alkaloids, glycosides, phenolic compounds, flavonoids, tannins, sterols, saponins, and terpenoids in PHF. 

### 2.2. Quantitative Estimation of Total Phenolic, Flavonoid, and Tannins Content in PHF

#### 2.2.1. Total Phenolic Content

The total phenolic content (TPC) of gallic acid and PHF was 968.42 ± 2.34 and 756.98 ± 3 ± 1.21 mg GAE/g of dry extract, respectively. PHF contains a high concentration of phenolic components.

#### 2.2.2. Total Flavonoid Content

Catechin and PHF had a total flavonoid content (TFC) of 976.48 ± 3.714 and 464.18 ± 4.32 mg CAE/g of dry extract, respectively. It indicates a high level of flavonoid content in PHF.

#### 2.2.3. Total Tannin Content

Catechin and PHF exhibited a total tannin content (TTC) of 477.26 ± 3.65 and 103.51 ± 4.84 mg CAE/g of dry extract, respectively. The quantity of tannins is lower in PHF.

### 2.3. Compatibility Study by Differential Scanning Calorimetry (DSC)

The DSC findings on metformin, polyherbal extract, and allopolyherbal formulation within the firstweek of their preparation are presented in Table 1 and Figure 1, Figure 2 and Figure 3. The DSC curves show that there is no overlapping, interference, or significant change in the peaks between metformin and extracts. As a result, metformin may be compatible with plant extracts.

### 2.4. Acute Toxicity Study of Polyherbal Formulation (PHF)

PHF did not show any drug-induced physical symptoms of toxicity, and no fatalities were recorded up to the dose of 2000 mg/kg body weight.

### 2.5. In Vivo Antidiabetic and Antihyperlipidemic Activity

Group I rats were considered normal and were simply given a vehicle (2.5% acacia arabica). Freshly prepared streptozotocin at a dosage of 55 mg/kg b.w was delivered intraperitoneally into group II and deemed as diabetic control.

Polyherbal formulations A, B, and C (PHF A, PHF B, and PHF C) were developed by combining CAr, CAs, and ZO in the following ratios: 1:1:1, 2:21, and 2:1:2. PHF A wasgiven to groups III and IV at doses of 200 mg/kg b.w. and 400 mg/kg b.w., respectively. PHF B was given to groups V and VI at doses of 200 and 400 mg/kg b.w., respectively. PHF C was administered to rats in groups VII (200 mg/kg b.w.) and VIII (400 mg/kg b.w.).

APHF wasmade by mixing PHF B (which is thought to be more significant than PHF A and C) with metformin in three distinct ratios. Groups IX, X, and XI received APHF A (PHF 200 mg/kg b.w + metformin 22.5 mg/kg b.w), APHF B (PHF 200 mg/kg b.w + metformin 45mg/kg b.w), and APHF C (PHF 200 mg/kg b.w + metformin 67.5mg/kg b.w). Group XII wasgiven a standard medication metformin at a dose of 90 mg/kg b.w.

All diabetic rats were given treatment for 21 consecutive days by oral approach. Variations in blood glucose levels and body weight were noted on the 1st, 7th, 14th, and 21st days of therapy. Total cholesterol (TC), triglycerides (TG), low density lipoproteins (LDL), very lowdensity lipoproteins (VLDL), highdensity lipoproteins (HDL), serum glutamate oxaloacetate transaminase (SGOT), serum glutamate pyruvate transaminase (SGPT), alkaline phosphatase (ALP), glycated hemoglobin (HbA1c), serum urea, creatinine, total protein, albumin, and total and differential white blood cell (WBC) count wereestimated on 21st day of treatment. The results are presented in Table 2, Table 3, Table 4, Table 5, Table 6 and Table 7 and Figure 4, Figure 5, Figure 6 and Figure 7. 

From a mechanistic perspective, the notable antidiabetic activity observed in this study can be ascribed to several pivotal factors. The sustained decrease in blood glucose levels over the course of treatment signifies enhanced glycemic control. It potentially arises from increased insulin secretion, restoration of beta cells within the pancreatic islets, improved glucose utilization by peripheral tissues, and inhibition of alpha-amylase and alpha-glucosidase enzymes. Additionally, the positive alterations in lipid profile parameters, such as decreased total cholesterol (TC), triglycerides (TG), LDL, and VLDL, along with an increase in HDL, suggest a potential mechanism for the formulation’s lipid-lowering effects. The normalization of liver enzyme levels (SGOT, SGPT, and ALP) reflects its hepatoprotective properties, while the reduction in HbA1c underscores its ability to manage long-term glucose regulation. Moreover, the maintenance of kidney function markers (serum urea and creatinine) and the preservation of total protein, albumin, and white blood cell counts indicate the formulation’s systemic and renoprotective influence. These comprehensive findings collectively support the polyherbal formulation’s multifaceted mechanisms for effectively managing diabetes and its associated complications.

#### 2.5.1. Effect on Blood Glucose Level

Table 2 and Figure 4 show the effects of PHF (A, B, and C), APHF (A, B, and C), and metformin on blood sugar levels in streptozotocin-induced diabetic rats. Blood glucose levels were determined in normal and diabetic rats on the 1st, 7th, 14th, and 21st days of therapy. Untreated diabetic rats had much higher glucose levels than control rats. Oral treatment of PHF A, B, and C at 200 and 400 mg/kg b.w, APHF A, B, and C, and the standard medicine metformin at 90 mg/kg b.w resulted in a substantial reduction in blood glucose levels (*p* < 0.05) in a dose- and time-dependent manner. APHF was the most active of the test samples, followed by PHF. Among all PHFs, PHF B had the most noticeable outcomes, while among APHFs, APHF C had the most prominent activity. As compared to metformin, APHF C was considerably more effective in lowering blood glucose levels.

#### 2.5.2. Effect on Bodyweight

Table 3 and Figure 5 show the differences in body weight between treatment groups. From day 1 to day 21, the diabetes control group’s body weight was dramatically lowered. Nonetheless, it is worth noting that the animals’ body weight was greatly restored after treatment with PHF, APHF, and metformin in a time- and dose-dependent manner.

#### 2.5.3. Effect on Lipid Profile

Table 4 and Figure 6 illustrate the effects of PHF (A, B, and C), APHF (A, B, and C), and metformin on the lipid profile in streptozotocin-induced diabetic rats. On the 21st day, diabetic control rats had significantly higher levels of TC, TG, LDL, VLDL, and lower levels of HDL. Similar changes in the lipid profile were avoided in the treatment groups and brought closer to normal dosage dependently. Metformin produced the best outcomes, followed by APHF and finally PHF. PHF B produced the most notable outcomes among all PHF, while APHF C produced the most notableactivity. Nonetheless, it is worth noting that the improvements in lipid profile are greater in APHF C than metformin.

#### 2.5.4. Effect on SGOT, SGPT, and ALP Levels

Table 5 and Figure 7 show the levels of blood enzymes such as SGOT, SGPT, and ALP in diabetic rats treated with PHF, APHF, and metformin. When diabetic control rats were compared to normal rats, the activity of these enzymes was dramatically elevated. On the 21st day of therapy, diabetic rats were given various doses of PHF, APHF, and metformin, and blood enzymes were considerably reduced in a dose-dependent manner.

The sequence of drug potency is APHF C > metformin > APHF B> APHFA > PHF B> PHF C > PHF A.

#### 2.5.5. Effect on Total and Differential White Blood Cell Count

The total and differential WBC counts of rats treated with PHF, APHF, and metformin were compared to those of normal and diabetic control groups of rats to determine the capacity of the medications to maintain normal WBC counts, and the findings are shown in Table 6.

Diabetic control rats had a significantly lower total leucocyte count. However, in the therapy group, the fall in WBC count was avoided. The WBC count was greater in all treatment groups when compared to the diabetes control group. Metformin improved WBC count recovery more than APHF and PHF. There were no significant differences in differential WBC count; however, the proportion of lymphocytes was reduced in diabetic control rats and prevented in treatment groups.

#### 2.5.6. Effect on HbA1c, Urea, Creatinine, Total Protein, and Albumin

Table 7 shows the effects of PHF (A, B, and C), APHF (A, B, and C), and metformin on HbA1c, urea, creatinine, total protein, and albumin. Diabetic rats’ serum contained significantly higher levels of urea, creatinine, and HbA1c, as well as lower levels of total protein and albumin. Serum urea level and HbA1c% decreased significantly in the therapy group. Serum creatinine levels are also reduced, but not dramatically. Total protein and albumin levels in the serum of the treated rat group increased as well, although these variations were not statistically significant.

PHF at 400 mg/kg b.w, metformin, and APHF exhibit a substantial rise in total protein (A, B, and C). Albumin levels increased significantly withAPHF B, C, and metformin.

#### 2.5.7. Histopathological Studies of Liver Cells

Normal control rats had a normal liver histoarchitecture. The hepatic parenchyma and sinusoids seemed normal, with normal Kupffer cell distribution. The livers of diabetic rats displayed necrotic changes, dilatation of the liver sinusoids, Kupffer cell activation, and cytoplasmic vacuolization of hepatocytes. Periportal inflammation, periportal infiltration, pykontic nuclei, and periportal fatty infiltration were also seen. Diabetic rats given metformin, PHF, or APHF had various degrees of augmentation. The liver of the metformin-treated group exhibited mild degenerative changes, moderate hepatic sinusoid dilatation, mild periportal inflammation, and a slightly greater number of Kupffer cells. The treated group’s liver histoarchitecture was more or less normal, with only minimal necrotic alterations and degeneration (Figure 8).

#### 2.5.8. Histopathological Studies of Pancreatic Cells

Normal control rats’ pancreases had normal histological architecture in the form of an acinar structure with normal beta cell islets. The number of Langerhans islets in the diabetic control group’s pancreases was significantly reduced. Beta cell depletion, dilated islets, acini shrinkage, and vacuolar degeneration were all seen. Pancreatic cell recovery and improvement were found in the PHF-, APHF-, and metformin-treated groups. PHF-treated rats’ pancreas revealed a moderate restoration of IL cells with fewnecrotic alterations. The pancreas of the APHF-treated group seemed normal, with restored islets and fewnecrotic alterations. The metformin-treated group showed restoration of islets to normal size, with normal acinar cells and fewnecrotic alterations (Figure 9).

## 3. Materials and Methods

### 3.1. Chemicals and Reagents

Streptozotocin and metformin were acquired from Yarrow Chem Products, Mumbai, Maharashtra, India. The remaining chemicals and reagents were of analyticalgrade.

### 3.2. Plant Material

Leaves of *Cassia auriculata* (CAr), *Centella asiatica* (CAs), and *Zingiber officinale* (ZO) rhizomes were obtained around Nagamangala Taluk, Mandya District, Karnataka, India. Plant samples were certified at the Foundation for Revitalisation of Local Health Traditions herbarium in Bangalore, India (FRLHT Acc. No. of plants is 5551, 5552, and 5553 for CAs, CAr, and ZO, respectively). The plant’s components were thoroughly cleaned, shade dried, and pulverized to get a coarse powder. Powders were kept in an airtight container in a sterile environment for later usage.

### 3.3. Preparation of Extract

The cold maceration method was followed to extract dried powders of CAr, CAs, and ZO using 70% ethanol [17,18].

### 3.4. Preparation of Polyherbal and Allopolyherbal Formulations

Polyherbal formulations were developed by combining CAr, CAs, and ZO in three distinct ratios: 1:1:1 (PHF A), 2:2:1 (PHF B), and 2:1:2 (PHF C), and were given to rats at doses of 200 and 400 mg/kg body weight. The APHF formulation was produced by combining a potent polyherbal formulation (PHF B) with metformin.

The following are the APHF specifics:APHF A: PHF B (200 mg/kg b.w) + 25% of metformin standard dose (22.5 mg/kg b.w)APHF B: PHF B (200 mg/kg b.w) + 50% of metformin standard dose (45 mg/kg b.w)APHF C: PHF B (200 mg/kg b.w) + 75% of metformin standard dose (67.5 mg/kg b.w)

### 3.5. Preliminary Phytochemical Investigation of Polyherbal Formulation

Qualitative phytochemical analysis of polyherbal formulations was carried out using standard procedures [19,20].

### 3.6. Quantitative Estimation of Total Phenolic, Flavonoid, and Tannin Content

The pathophysiology of many acute and chronic disorders is influenced by oxidative stress. Free radicals and other reactive oxygen species (ROS) have been linked to the pathophysiology of a variety of diseases, including diabetes. Antioxidants such as phenols, flavonoids, and tannins scavenge free radicals such as peroxide, hydroperoxide, and lipid peroxyl, which delay or prevent oxidative damage to the target molecule [21,22].

#### 3.6.1. Estimation of Total Phenolic Content

The total phenolic content of PHF was determined using the Folin–Ciocâlteu (FC) reagent. The PHF was combined in a specific volume with FC reagent (10 times pre-diluted). Later, 1.6 mL of a 7.5% *w*/*v* sodium carbonate solution was added. The solution was stirred for 1 h before being incubated at room temperature. The absorbance at 765 nm was measured using a UV–visible spectrophotometer (Shimadzu UV-1700, PharmaSpec (Metro Manila, Philippines). A calibration curve for the standard solution (gallic acid) was created [23,24].

#### 3.6.2. Estimation of Total Flavonoid Content

The total flavonoid content was determined colorimetrically using the aluminum chloride procedure. A known volume of the PHF or catechin was mixed with distilled water (1.25 mL) in a test tube, and a 5% sodium nitrite solution (75 µL) was added. Following that, 150 µL of a 10% aluminum chloride solution was added and left to rest for 5 min before adding 1 M sodium hydroxide (0.5 mL). The absorbance at 510 nm was measured immediately using a UV–visible spectrophotometer. The findings were expressed in milligrams of (+)-catechin equivalent per gram of extract. The calibration curve ofcatechin (standard reference drug) was drawn [25,26].

#### 3.6.3. Estimation of Total Tannin Content

The vanillin–HCl procedure wasused to determine the total tannin content.

First, 250 µL of PHF was treated with 2250 µL of reagent combination (4:1 ratio of 4% vanillin and 8% concentrated HCl produced in methanol). For around 20 min, the resulting mixture was incubated at room temperature. At 500 nm, the absorbance was measured. The calibration curve was created using various concentrations of catechin (20–100 µg/mL), and the total tannin value was reported as mg catechin equivalents (CAE)/g dry extract [24].

### 3.7. Compatibility Study of APHF by Differential Scanning Calorimetry (DSC)

A compatibility study is one of the most vital areas of ensuring the stability of an allopolyherbal formulation and revealing potential interactions that may arise between the components in a mixture [27]. An in-depth understanding of the physical interactions between the components in a formulation is feasible with a thermal (differential scanning calorimetry) approach [28]. The compatibility of metformin with polyherbal extract wasinvestigated using DSC. This study looked for any potential physical interactions between the extract and metformin once they weremixed.

The samples were analyzed using Shimadzu heat-flux DSC in the temperature range of 0 °C to 600 °C, with a heating rate of 6 K/min and in an environment of flowing nitrogen [29].

### 3.8. InVivo Antidiabetic Activity

#### 3.8.1. Experimental Animals

Albino rats weighing around 160–200 g and aged 3 to 4 months were obtained from Vaarunya Biolabs Pvt. Ltd., Bengaluru, India (CPCSEA Registration No. 2076/P O/RcBiBt/S/19/C PCSEA), Bengaluru, India. Rats were housed in spacious polyacrylic cages under typical husbandry conditions (25 °C temperature, 60–70% relative humidity, and a 12-h light/dark cycle). They were fed regular rat pellets and had unlimited access to drinking water. Experiments were carried out with the approval of Institutional Animal Ethical Committee (IAEC) of Sri Adichunchanagiri, College of Pharmacy, India (IAEC Approval No. SACCP-IAEC/2021-02/49 and SACCP-IAEC/2022-01/58).

#### 3.8.2. Acute Toxicity Study of Polyherbal Formulation

Individually, all of the extracts employed in the development of the polyherbal formulation were shown to be safe [14,30,31]. As a result, an acute toxicity study using a limit test at one dose level of 2000 mg/kg body weight with six rats (three rats per step) for 14 days in each step was performed for the polyherbal extract (made by mixing equal proportions of CAr, CAs, and ZO).

#### 3.8.3. Induction of Diabetes

To induce diabetes in overnight starved rats, a single intraperitoneal injection of freshly prepared STZ (55 mg/Kg b.w, Yarow Chem Products, Mumbai, Maharashtra, India) in 0.1 M citrate buffer (pH 4.5, Sisco Research Laboratories Pvt. Ltd., Taloja, Maharashtra, India) was employed. To prevent STZ-induced hypoglycemic death, the rats were given a 20% glucose solution to drink for 24 h. Diabetes was diagnosed one week after the STZ injection, and rats with fasting blood glucose levels of greater than 200 mg/dL were chosen for the investigation [32,33].

#### 3.8.4. Experimental Design

The rats (n = 72) were divided into twelve groups, each comprising six rats. The test compounds were suspended in distilled water (dH_2_O) in a vehicle containing 2.5% acacia suspension (Yarow Chem Products, Mumbai, Maharashtra, India) and administered orally once a day for 21 days to all rats. The anti-diabetic effects of individual extracts, PHF, and metformin were first tested. Based on the results, a very effective PHF ratio was chosen for the development of APHF. The antidiabetic activity of APHF was tested later.

Rats were grouped as follows:Group I: Served as a normal control (2.5% acacia suspension b.w, p.o)Group II: Served as a diabetic control (Streptozotocin 55 mg/kg b.w, i.p)Group III: Treatment with PHF A—1:1:1 (200 mg/kg b.w, p.o)Group IV: Treatment with PHF A—1:1:1 (400 mg/kg b.w, p.o)Group V: Treatment with PHF B—2:2:1 (200 mg/kg b.w, p.o)Group VI: Treatment with PHF B—2:2:1 (400 mg/kg b.w, p.o)Group VII: Treatment with PHF C—2:1:2 (200 mg/kg b.w, p.o)Group VIII: Treatment with PHF C—2:1:2 (400 mg/kg b.w, p.o)Group IX: APHF A (200 mg/kg PHF B+22.5 mg/kg metformin b.w, p.o)Group X: APHF B (200 mg/kg PHF B+45 mg/kg metformin b.w, p.o)Group XI: APHF C (200 mg/kg PHF B+67.5 mg/kg metformin b.w, p.oGroup XII: Treatment with metformin (90 mg/kg b.w, p.o)

#### 3.8.5. Body Weight (b.w) Analysis

The b.w of each rat was measured using an animal weighing balance (Docbel Industries, New Delhi, India) at 7-day intervals from the start to the completion of the experiment (i.e., the 1st, 7th, 14th, and 21st days).

#### 3.8.6. Blood Glucose Level

Metformin, PHF, and APHF were administered to diabetic rats for 21 days. Blood samples were taken by rupturing the tail vein, and blood glucose was determined on the 1st, 7th, 14th, and 21st days of therapy using an On Call Plus glucometer (ACON Biotech (Hangzhou) Co., Ltd., West Lake District, Hangzhou, China). 

#### 3.8.7. Biochemical Analysis

The rats were sacrificed under anesthetic conditions (ketamine hydrochloride, 80 mg/kg b.w, intraperitoneally) at the end of the experiment (on the 21st day). Blood samples were collected through cardiac puncture and analyzed in accordance with previously reported procedures [34,35,36]. Lipid profiles such as total cholesterol (TC), triglycerides (TG), high-density lipoprotein (HDL), low-density lipoprotein (LDL), and very low-density lipoprotein (VLDL) were assessed. Serum glutamate oxaloacetate transaminase (SGOT), serum glutamate pyruvate transaminase (SGPT), alkaline phosphatase (ALP), glycated hemoglobin (Hb1Ac), urea, creatinine, total protein, albumin, and total count and differential count of leucocyte were also estimated.

#### 3.8.8. Histopathological Analysis

The pancreas and liver were excised, rinsed constantly with phosphate buffered saline (1 PBS, pH 7.4), preserved in a 10% formalin solution, and embedded in paraffin blocks. Five micrometer thick slices were cut on a semi-automated rotator microtome and stained with hematoxylin and eosin. An inverted biological microscope (FM-BM-B200, Fison Instruments Ltd., Glasgow G2 4JR, UK) was used to image the sections [37]. 

#### 3.8.9. Statistical Analysis

Results were presented as mean SEM, and statistical differences between groups were established using analysis of variance (ANOVA, GraphPad Prism version 8.0.2, San Diego, CA, USA). Statistical significance was defined as *p*-values less than 0.05. 

## 4. Conclusions

The biguanide class of drugs (metformin) has been linked to a variety of side effects, including metallic taste, nausea, vomiting, diarrhea, anorexia, and lactic acidosis. The dose of marketed metformin used by patients is greater when compared to other allopathic hypoglycemic medications. The suggested research has the potential to minimize unwanted side effects and/or decrease the dose of allopathic medicine, specifically metformin, by substituting it with a polyherbal formulation (PHF) and/or an allopolyherbal formulation (APHF). PHF and APHF both showed synergistic anti-diabetic efficacy. The anti-diabetic impact of APHF is stronger than that of PHF.

In comparison to metformin alone at higher doses, APHF and a relatively low dose of metformin together produce an extremely potent hypoglycemic effect. As a result, the current study shows that PHF and APHF are safe and effective medications in the treatment of diabetes mellitus because they serve to replace or reduce the dose of metformin, hence lowering metformin hazards. Further research is needed to confirm the safety and efficacy of APHF in human volunteers, followed by its formulation into an acceptable dosage form that piques the interest of patients.

## Figures and Tables

**Figure 1 pharmaceuticals-16-01368-f001:**
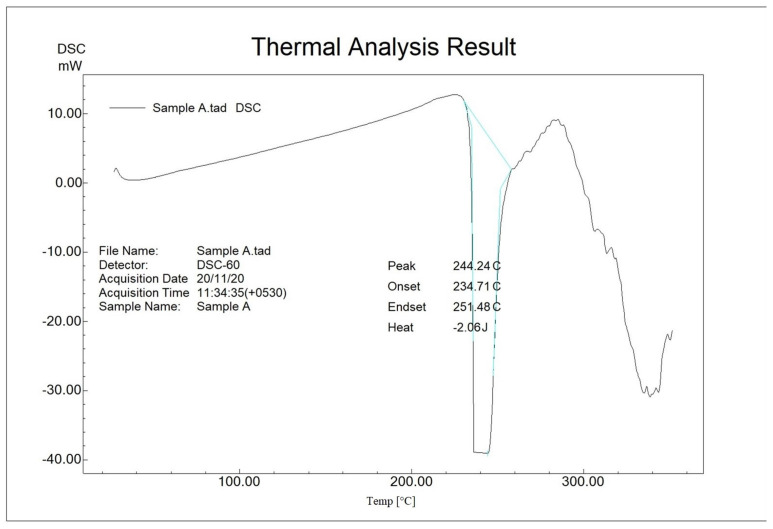
DSC curve of metformin.

**Figure 2 pharmaceuticals-16-01368-f002:**
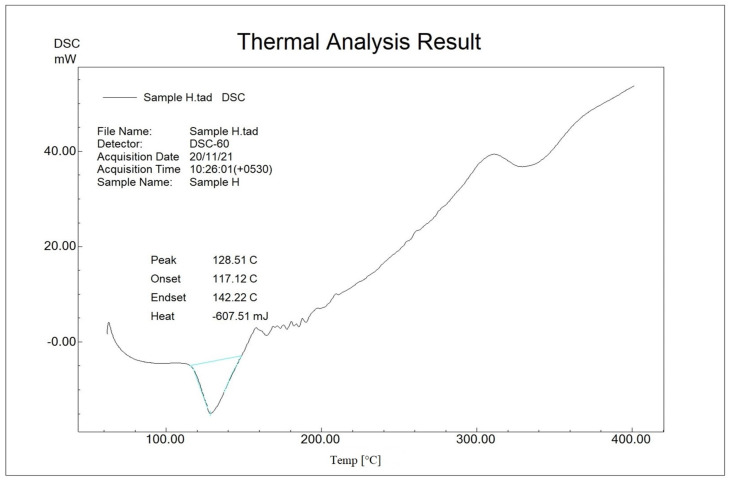
DSC curve of polyherbal extract.

**Figure 3 pharmaceuticals-16-01368-f003:**
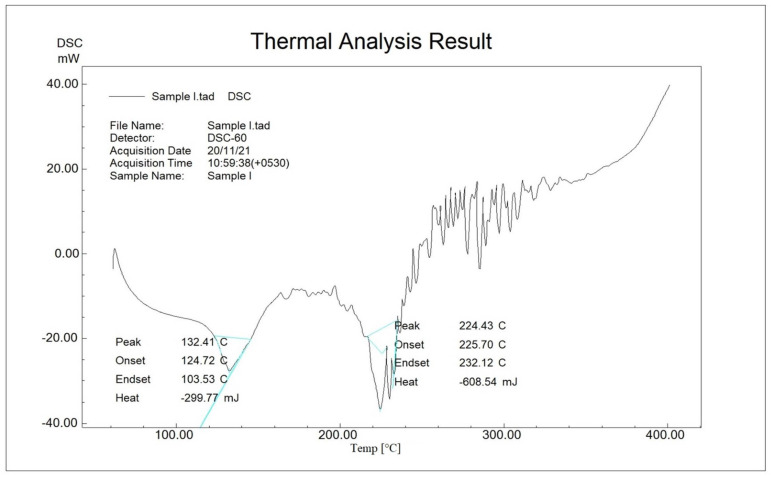
DSC curve of allopolyherbal extract.

**Figure 4 pharmaceuticals-16-01368-f004:**
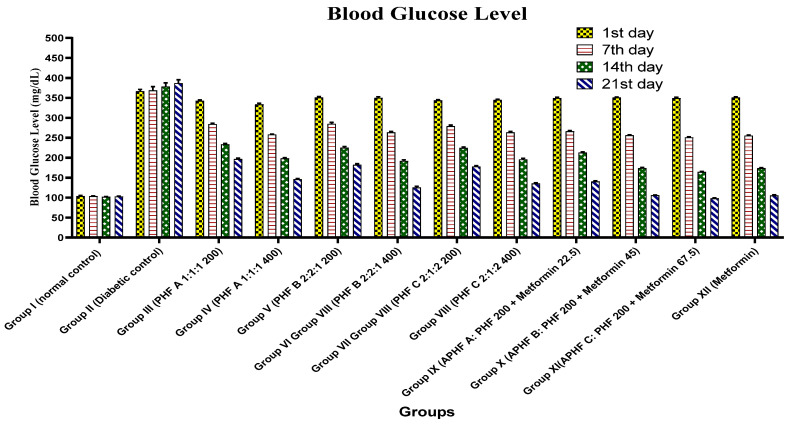
Effect of drugs on blood glucose level in STZ-induced diabetic rats (metformin vs. PHF vs. APHF).

**Figure 5 pharmaceuticals-16-01368-f005:**
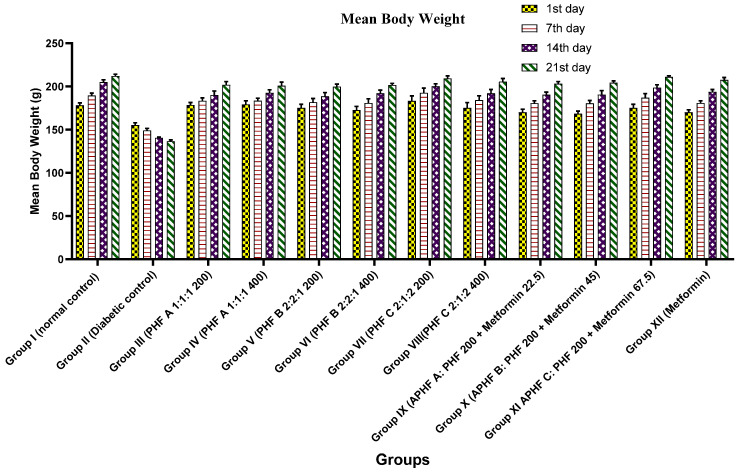
Effect of drugs on body weight (g) in STZ-induced diabetic rats (metformin vs. PHF vs. APHF).

**Figure 6 pharmaceuticals-16-01368-f006:**
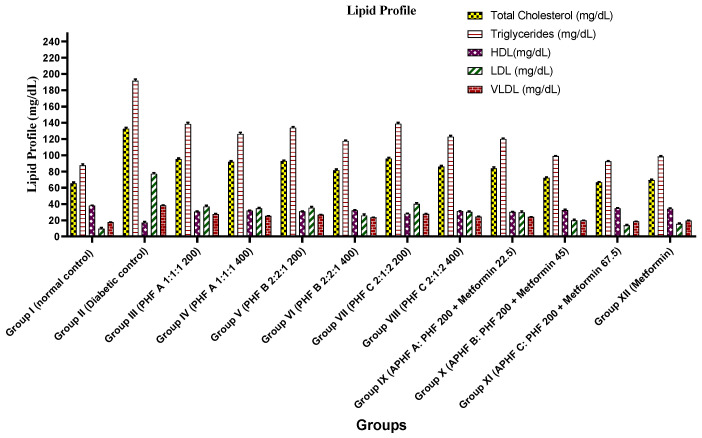
Effect of drugs on lipid profile in STZ-induced diabetic rats (metformin vs. PHF vs. APHF).

**Figure 7 pharmaceuticals-16-01368-f007:**
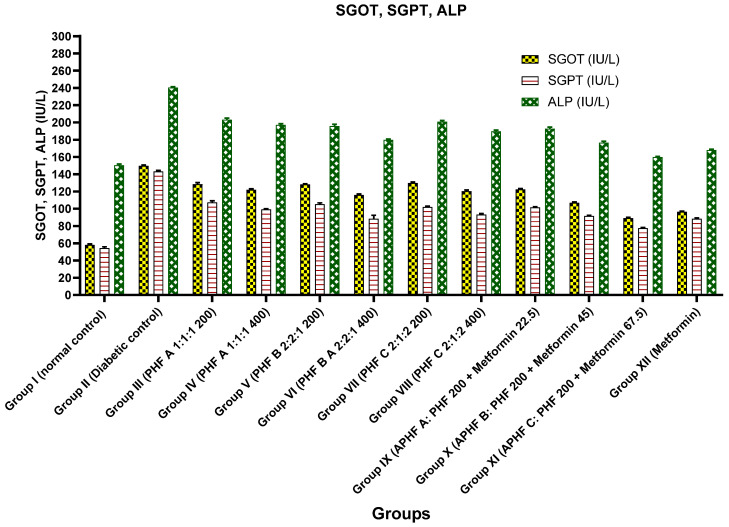
Effect of drugs on SGPT, SGOT, and ALP levels in STZ-induced diabetic rats (metformin vs. PHF vs. APHF).

**Figure 8 pharmaceuticals-16-01368-f008:**
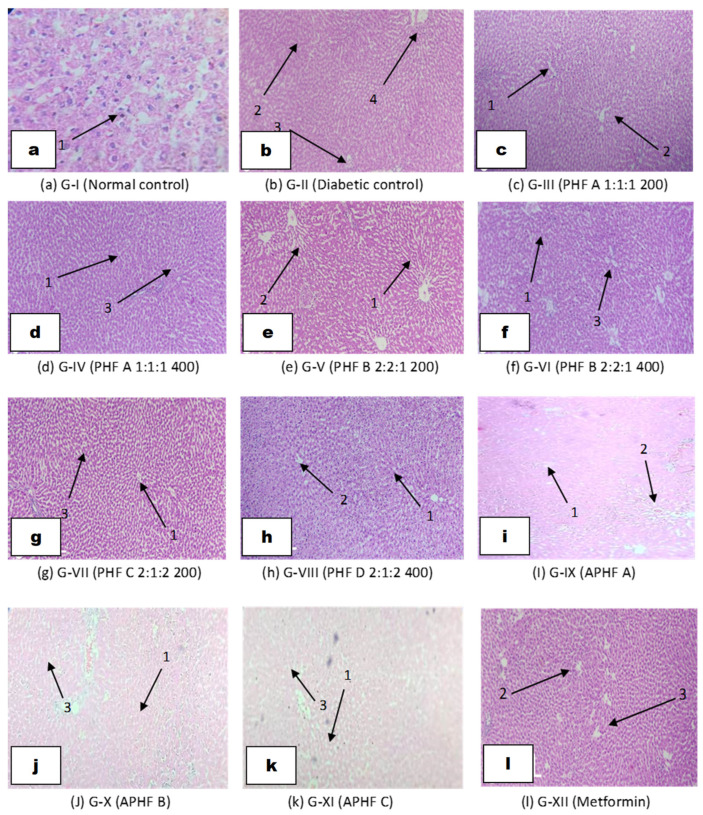
The liver section of histology. (**a**) G-I (normal control), (**b**) G-II (diabetic control), (**c**) G-III (PHF A 1:1:1 200), (**d**) G-IV (PHF A 1:1:1 400), (**e**) G-V (PHF B 2:2:1 200), (**f**) G-VI (PHF B 2:2:1 400), (**g**) G-VII (PHF C 2:1:2 200), (**h**) G-VIII (PHF C 2:1:2 400), (**i**) G-IX (APHF A), (**j**) G-X (APHF B), (**k**) G-XI (APHF C), (**l**) G-XII (metformin),where 1: normal hepatocyte, 2: pykontic nuclei, 3: mild periportal inflammation, 4: periportal fatty infiltration.

**Figure 9 pharmaceuticals-16-01368-f009:**
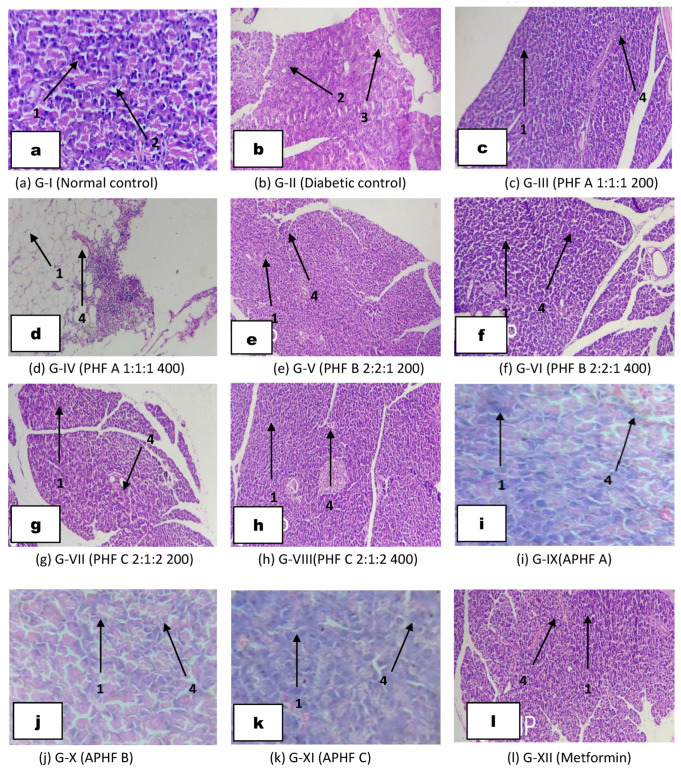
The pancreas section of histology. (**a**) G-I (normal control), (**b**) G-II (diabetic control), (**c**) G-III (PHF A 1:1:1 200), (**d**) G-IV (PHF A 1:1:1 400), (**e**) G-V (PHF B 2:2:1 200), (**f**) G-VI (PHF B 2:2:1 400), (**g**) G-VII (PHF2:1:2 200), (**h**) G-VIII (PHF C 2:1:2 400), (**i**) G-IX (APHF A), (**j**) G-X (APHF B), (**k**) G-XI (APHF C), (**l**) G-XII (metformin),where 1: beta cells islets, 2: depletion of beta cells, 3: deleted islets, 4: formation of islets.

**Table 1 pharmaceuticals-16-01368-t001:** DSC data of polyherbal extract and allopolyherbal formulation.

SL. NO	MATERIALS	PEAK (°C)	ONSET (°C)	ENDSET (°C)	HEAT (mJ)
1	Metformin (Figure 1)	244.2	234.7	251.4	−2.0
2	Polyherbal extract (Figure 2)	128.5	117.1	142.2	−607.5
3	Allopolyherbal formulation (Polyherbal + metformin) (Figure 3)	132.4 224.4	124.7 225.7	103.5 232.1	−299.7 −608.5

**Table 2 pharmaceuticals-16-01368-t002:** Effect of drugs on blood glucose level in STZ-induced diabetic rats (metformin vs. PHF vs. APHF).

Group	Blood Glucose Levels (mg/dL) (Mean ± SEM)			
	1st Day	7th Day	14th Day	21st Day
Group I (Normal control)	103.8 ± 1.6	103.6 ± 0.7	102.0 ± 0.9	103.0 ± 1.0
Group II (Diabetic control)	366.6 ± 4.6	368.8 ± 9.5	378.3 ± 9.2	386.6 ± 8.8
Group III (PHF A 1:1:1 200)	342.5 ± 2.6 *	284.5 ± 2.1 ***	233.8 ± 2.1 ***	196.6 ± 2.4 ***
Group IV (PHF A 1:1:1 400)	333.6 ± 3.0 *	258.1 ± 1 ***	198.5 ± 1.8 ***	146.0 ± 2.0 ***
Group V (PHF B 2:2:1 200)	351.0 ± 2.5 *	285.0 ± 3.5 ***	225.0 ± 3.2 ***	182.1 ± 2.8 ***
Group VI (PHF B 2:2:1 400)	350.1 ± 2.4 *	263.5 ± 2.07 ***	191.8 ± 2.9 ***	125.3 ± 3.2 ***
Group VII (PHF C 2:1:2 200)	343.5 ± 1.7 *	279.1 ± 2.7 ***	224.5 ± 2.0 ***	178.0 ± 2.3 ***
Group VIII (PHF C 2:1:2 400)	345.0 ± 1.7 *	263.5 ± 2.1 ***	195.8 ± 2.8 ***	135.5 ± 2.1 ***
Group IX APHF A	349.6 ± 1.8 *	266.3 ± 2.0 ***	212.6 ± 1.8 ***	140.3 ± 2.2 ***
Group X APHF B	351 ± 1.1 *	256.1 ± 1.3 ***	174.1 ± 1.3 ***	105.8 ± 0.9 ***
Group XI APHF C	349.3 ± 2.2 *	250.8 ± 1.4 ***	164.3 ± 1.5 ***	98.3 ± 1.3 ***
Group XII (Metformin)	350.8 ± 1.7 *	255.5 ± 1.1 ***	173.8 ± 1.49 ***	105.3 ± 1.4 ***

Values are expressed as mean ± SEM, (n = 6); *p*< 0.05 (*), *p* < 0.001 (***) compared to diabetic animals (two-way ANOVA followed by a Dunnett’s *t*-test). *p*-values < 0.05 were considered statistically significant.

**Table 3 pharmaceuticals-16-01368-t003:** Effect of drugs on body weight (g) in STZ-induced diabetic rats (metformin vs. PHF vs. APHF).

Group	Mean Body Weight (g) (Mean ± SEM)			
	1st Day	7th Day	14th Day	21st Day
Group I (Normal control)	178.1 ± 2.7	189.5 ± 2.7	205 ± 2.6 ***	212 ± 2.0 ***
Group II (Diabetic control)	155.1 ± 2.6	149 ± 2.3	140.5 ± 0.8 ***	136.5 ± 1.5 ***
Group III (PHF A 1:1:1 200)	178.3 ± 3.0 ***	183.3 ± 3.3 ***	190 ± 4.6 ***	201.8 ± 3.7 ***
Group IV (PHF A 1:1:1 400)	179.1 ± 4.1 ***	183.5 ± 2.8 ***	192.5 ± 3.5 ***	200.8 ± 4.1 ***
Group V (PHF B 2:2:1 200)	175 ± 4.2 **	181.6 ± 4.3 ***	188.5 ± 4.3 ***	199.6 ± 2.9 ***
Group VI (PHF B 2:2:1 400)	172.5 ± 4.4 **	180.8 ± 4.8 ***	192.1 ± 3.6 ***	201.6 ± 1.8 ***
Group VII (PHF C 2:1:2 200)	183.1 ± 5.8 ***	192.5 ± 5.4 ***	200.1 ± 2.8 ***	209.1 ± 3.0 ***
Group VIII (PHF C 2:1:2 400)	175 ± 6.1 **	184.1 ± 5.0 ***	192 ± 4.4 ***	205.5 ± 3.6 ***
Group IX APHF A	170 ± 3.6 **	180.5 ± 2.8 ***	190.3 ± 3.0 ***	203 ± 2.5 ***
Group X APHF B	168.3 ± 3.0 **	180.3 ± 3.4 ***	190.5 ± 4.6 ***	204.3 ± 2.0 ***
Group XI APHF C	175 ± 4.2 **	187 ± 4.6 ***	198.5 ± 3.4 ***	211 ± 1.3 ***
Group XII (Metformin)	170 ± 2.8 *	180.8 ± 2.5 ***	193.6 ± 2.7 ***	207.5 ± 2.8 ***

Values are expressed as mean ± SEM, (n = 6); *p* < 0.05 (*), *p* < 0.01 (**), *p* < 0.001 (***) compared to diabetic animals (two-way ANOVA followed by a Dunnett’s *t*-test). *p*-values < 0.05 were considered statistically significant.

**Table 4 pharmaceuticals-16-01368-t004:** Effect of drugs on lipid profile in STZ-induced diabetic rats (metformin vs. PHF vs. APHF).

Group	Lipid Profile (mg/dL)				
	TC	TG	HDL	LDL	VLDL
Group I (Normal control)	65.3 ± 1.5 ***	87.6 ± 1.7 ***	38 ± 0.3 ***	9.8 ± 1.1 ***	17.5 ± 0.3 ***
Group II (Diabetic control)	132.8 ± 1.3 ***	191.6 ± 2.0 ***	17.1 ± 1.1 ***	77.3 ± 1.2 ***	38.3 ± 0.4 ***
Group III (PHF A 1:1:1 200)	95.3 ± 1.3 ***	138.6 ± 1.6 ***	30.6 ± 0.4 ***	36.9 ± 1.3 ***	27.7 ± 0.3 ***
Group IV (PHF A 1:1:1 400)	91.8 ± 1.3 ***	126.1 ± 1.9 ***	32 ± 0.3 ***	34.6 ± 1.1 ***	25.2 ± 0.3 ***
Group V (PHF B 2:2:1 200)	93 ± 0.9 ***	134 ± 1.3 ***	30.8 ± 0.6 ***	35.3 ± 1.4 ***	26.8 ± 0.2 ***
Group VI (PHF B 2:2:1 400)	81.6 ± 1.3 ***	117.6 ± 0.7 ***	32.3 ± 0.5 ***	25.8 ± 1.8 ***	23.5 ± 0.1 ***
Group VII (PHF C 2:1:2200)	96 ± 1.2 ***	139 ± 1.3 ***	28 ± 0.6 ***	40.2 ± 1.3 ***	27.8 ± 0.2 ***
Group VIII (PHF C 2:1:2400)	86.1 ± 1.1 ***	122.8 ± 1.4 ***	31.3 ± 0.4 ***	30.2 ± 0.8 ***	24.5 ± 0.2 ***
Group IX APHF A	83.8 ± 1.6 ***	120 ± 0.9 ***	30.1 ± 0.4 ***	29.6 ± 1.8 ***	24 ± 0.1 ***
Group XAPHF B	72.3 ± 0.8 ***	99 ± 0.5 ***	32.5 ± 1.0 ***	20.0 ± 1.0 ***	19.8 ± 0.1 ***
Group XI APHF C	66.8 ± 0.6 ***	92.5 ± 0.7 ***	34.6 ± 0.7 ***	13.6 ± 1.0 ***	18.5 ± 0.1 ***
Group XII (Metformin)	69.5 ± 0.9 ***	98.3 ± 0.8 ***	34.1 ± 0.7 ***	15.6 ± 1.0 ***	19.6 ± 0.1 ***

Values are expressed as mean ± SEM, (n = 6); *p* < 0.001 (***) compared to diabetic animals (two-way ANOVA followed by a Dunnett’s *t*-test). *p*-values < 0.05 were considered statistically significant.

**Table 5 pharmaceuticals-16-01368-t005:** Effect of drugs on SGPT, SGOT andALP levels in STZ-induced diabetic rats (metformin vs. PHF vs. APHF).

Group	SGOT Level (IU/L)	SGPT Level (IU/L)	ALP Level (IU/L)
Group I (Normal control)	58 ± 1.2	54.3 ± 1.5	150.3 ± 1.3
Group II (Diabetic control)	149.8 ± 0.7	143.1 ± 1.2	240.3 ± 0.8
Group III (PHF A 1:1:1 200)	128.3 ± 1.9 ***	107.1 ± 2.0 ***	203.1 ± 1.8 ***
Group IV (PHF A 1:1:1 400)	121.8 ± 1.2 ***	99.2 ± 0.8 ***	197.1 ± 1.4 ***
Group V (PHF B 2:2:1 200)	128.1 ± 0.7 ***	105.3 ± 1.4 ***	195.8 ± 1.9 ***
Group VI (PHF B 2:2:1 400)	115.8 ± 1.1 ***	88.3 ± 4.1 ***	179.8 ± 0.9 ***
Group VII (PHF C 2:1:2 200)	129.8 ± 1.0 ***	102 ± 1.0 ***	200.8 ± 1.3 ***
Group VIII (PHF C 2:1:2 400)	120.3 ± 1.3 ***	93.1 ± 1.2 ***	189.6 ± 1.4 ***
Group IX APHF A	122.3 ± 1.1 ***	101.5 ± 0.9 ***	192.8 ± 1.7 ***
Group X APHF B	106.8 ± 1.1 ***	91.6 ± 0.6 ***	176.5 ± 1.6 ***
Group XI APHF C	89 ± 1.1 ***	77.5 ± 0.7 ***	159.8 ± 0.7 ***
Group XII (Metformin)	96.4 ± 0.9 ***	88.3 ± 1.0 ***	168 ± 0.9 ***

Values are expressed as mean ± SEM, (n = 6); *p* < 0.001 (***) compared to diabetic animals (two-way ANOVA followed by a Dunnett’s *t*-test). *p*-values < 0.05 were considered statistically significant.

**Table 6 pharmaceuticals-16-01368-t006:** Effect of drugs on leucocyte indices in STZ-induced diabetic rats (metformin vs. PHF vs. APHF).

Group	Leucocytes 10^3^/Cumm	Neutrophils (%)	Lymphocytes (%)	Monocytes (%)	Eosinophils (%)
Group I (Normal control)	10.0 ± 0.1	61.1 ± 0.4	33.1 ± 0.6	4.1 ± 0.3	1.5 ± 0.4
Group II (Diabetic control)	5.2 ± 0.1	63.8 ± 0.5	29.8 ± 0.7	4.5 ± 0.4	1.8 ± 0.3
Group III (PHF A 1:1:1 200)	6.7 ± 0.1 ^ns^	62 ± 0.8 ^ns^	32 ± 0.6 *	4.3 ± 0.3 ^ns^	1.6 ± 0.3 ^ns^
Group IV (PHF A 1:1:1 400)	7.8 ± 0.1 **	62 ± 0.9 ^ns^	32.5 ± 0.8 **	4.3 ± 0.3 ^ns^	1.1 ± 0.3 ^ns^
Group V (PHF B 2:2:1 200)	6.4 ± 0.1 ^ns^	62.1 ± 0.7 ^ns^	32.1 ± 0.7 *	4.1 ± 0.4 ^ns^	1.5 ± 0.5 ^ns^
Group VI (PHF B 2:2:1 400)	7.7 ± 0.1 **	61.5 ± 0.6 *	32.5 ± 0.7 **	4.3 ± 0.3 ^ns^	1.6 ± 0.4 ^ns^
Group VII (PHF C 2:1:2 200)	6.5 ± 0.0 ^ns^	62.1 ± 0.8 ^ns^	32.3 ± 0.9 **	4.1 ± 0.3 ^ns^	1.3 ± 0.4 ^ns^
Group VIII (PHF C 2:1:2 400)	7.9 ± 0.0 **	61.5 ± 0.4 *	33.3 ± 0.7 ***	4.1 ± 0.4 ^ns^	1 ± 0.4 ^ns^
Group IX APHF A	6.8 ± 0.0 ^ns^	61.1 ± 0.4 **	34 ± 0.2 ***	4.1 ± 0.4 ^ns^	0.6 ± 0.2 ^ns^
Group X APHF B	7.1 ± 0.0	62.1 ± 0.4 ^ns^	32.5 ± 0.6 ***	4.1 ± 0.5 ^ns^	1.1 ± 0.4 ^ns^
Group XI APHF C	7.9 ± 0.1 **	62.5 ± 0.7 ^ns^	32 ± 0.6 ***	4 ± 0.5 ^ns^	1.5 ± 0.4 ^ns^
Group XII (Metformin)	8.0 ± 0.0 ***	62.3 ± 0.9 ^ns^	32 ± 0.6 *	4.1 ± 0.4 ^ns^	1.5 ± 0.4 ^ns^

Values are expressed as mean ± SEM, (n = 6); *p* < 0.05 (*), *p* < 0.01 (**), *p* < 0.001 (***), ns (non-significant) compared to diabetic animals (two-way ANOVA followed by a Dunnett’s *t*-test). *p*-values < 0.05 were considered statistically significant.

**Table 7 pharmaceuticals-16-01368-t007:** Effect of drugs on HbA1c, serum urea, creatinine, total protein, and albumin in STZ-induced diabetic rats (metformin vs. PHF vs. APHF).

Group	HbA1c (%)	Urea (mg/dL)	Creatinine (mg/dL)	Total Protein (g/dL)	Albumin (mg/dL)
Group I (Normal control)	5.2 ± 0.0	29.3 ± 1.1	0.8 ± 0.0	7.3 ± 0.1	5.4 ± 0.0
Group II (Diabetic control)	15.0 ± 0.3	47.3 ± 0.6	2.0 ± 0.0	4.5 ± 0.0	2.9 ± 0.0
Group III (PHF A 1:1:1 200)	8.4 ± 0.0 ***	36.3 ± 1.0 ***	1.1 ± 0.0 ^ns^	6.1 ± 0.0 *	3.7 ± 0.0 ^ns^
Group IV (PHF A 1:1:1 400)	6.7 ± 0.0 ***	36.3 ± 0.8 ***	0.9 ± 0.0 ^ns^	6.5 ± 0.0 **	4.0 ± 0.0 ^ns^
Group V (PHF B 2:2:1 200)	7.9 ± 0.0 ***	38.1 ± 0.5 ***	1.2 ± 0.0 ^ns^	6.2 ± 0.0 *	4.0 ± 0.4 ^ns^
Group VI (PHF B 2:2:1 400)	5.9 ± 0.1 ***	34.1 ± 1.0 ***	1.1 ± 0.0 ^ns^	6.4 ± 0.0 **	4.1 ± 0.0 ^ns^
Group VII (PHF C 2:1:2 200)	7.8 ± 0.0 ***	38.6 ± 0.8 ***	1.2 ± 0.0 ^ns^	5.8 ± 0.0 ^ns^	3.8 ± 0.0 ^ns^
Group VIII (PHF C 2:1:2 400)	6.3 ± 0.0 ***	32.3 ± 0.6 ***	1.0 ± 0.0 ^ns^	6.1 ± 0.0 *	4.0 ± 0.0 ^ns^
Group IX APHF A	6.5 ± 0.0 ***	33.1 ± 0.9 ***	1.0 ± 0.0 ^ns^	6.1 ± 0.0 *	4.1 ± 0.0 ^ns^
Group X APHF B	5.3 ± 0.0 ***	34.5 ± 0.7 ***	0.9 ± 0.0 ^ns^	6.3 ± 0.0 **	4.4 ± 0.0 *
Group XI APHF C	5.0 ± 0.0 ***	29.8 ± 1.0 ***	0.8 ± 0.0 ^ns^	6.8 ± 0.0 ***	4.8 ± 0.0 **
Group XII (Metformin)	5.2 ± 0.0 ***	30.3 ± 0.8 ***	0.9 ± 0.0 ^ns^	6.7 ± 0.0 **	4.6 ± 0.1 *

Values are expressed as mean ± SEM, (n = 6); *p* < 0.05 (*), *p* < 0.01 (**), *p* < 0.001 (***), ns (non-significant) compared to diabetic animals (two-way ANOVA followed by a Dunnett’s *t*-test). *p*-values < 0.05 were considered statistically significant.

## Data Availability

Data that support the findings of this study are available from the corresponding author upon reasonable request.

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
