# Peer review of "Synergistic Antihyperglycemic and Antihyperlipidemic Effect of Polyherbal and Allopolyherbal Formulation"

_pharmaceuticals, 2023, doi:10.3390/ph16101368_

Round 1

Reviewer 1 Report

This is  an interesting manuscript. However, I have a few suggestions for improvement. Authors need to address the research gap. Because i don't find any thing unique in this study . Authors need to justify it, The authors need to explain why they have selected only two doses,200 and 400 mg/kg b.w: why not low, medium, and high doses? How about the LD50 of oral toxic dose? Author should enlight on this in their manuscript. The authors need to explain more clearly how the formulation doses were prepared. Overall, the parameters of the method chosen were satisfactory. The results obtained from the research are satisfactory. Discussion is needs to be improved. Authors should discuss their findings from a mechanistic perspective.               This is  an interesting manuscript. However, I have a few suggestions for improvement. Authors need to address the research gap. Because i don't find any thing unique in this study . Authors need to justify it, The authors need to explain why they have selected only two doses,200 and 400 mg/kg b.w: why not low, medium, and high doses? How about the LD50 of oral toxic dose? Author should enlight on this in their manuscript. The authors need to explain more clearly how the formulation doses were prepared. Overall, the parameters of the method chosen were satisfactory. The results obtained from the research are satisfactory. Discussion is needs to be improved. Authors should discuss their findings from a mechanistic perspective.          

Author Response

Authors are highly thankful to esteemed reviewers for their excellent comments for further improvement of the article.

We hope the important queries rose by esteemed reviewer # 1 have been answered and revised for their satisfaction. The comments raised by the reviewers have been answered/justified and listed point-by-point below and the corresponding revisions/corrections are highlighted in the revised Manuscript. 

Kindly consider and do the needful.

Thanks and regards,

Dr. Mohamed Rahamathulla

Reviewer 2 Report

I read the paper carefully and realized that it needs a major revision. The title of this paper sounds good, However, some ambiguous points within this submission should be addressed, modified, or clarified in its revised form. My general remarks are as follows:

1.     I suggest to shorten the background section of the abstract “Polyherbal formulation (PHF) boosts the therapeutic effect and also decreases undesirable side effects by reducing the dose of individual herbs. Allopolyherbal formulation (APHF) is a combination of allopathic medication with polyherbal extract. The majority of allopathic medications have one or more side effects. As a result, combining Polyherbs with allopathic drugs helps to re- duce the dose of allopathic drugs thereby eliminates or lowers adverse effects associated with it.” To be clear for readers.

2.     In the abstract the authors mentioned “PHF, APHF and metformin has been administered to albino rats for 21 consecutive days” kindly mention in this sentence the used treatment dose per/kg.

3.     In the introduction section, the authors mentioned “According to the World Health Organization (WHO), 171 million people were diagnosed with diabetes world- wide in 2000, with that figure expected to climb to 366 million by 2030” Replace this sentence with the current WHO last statistics to demonstrate clearly the problem statement.

4.     The English style of your manuscript is acceptable at a glance. However, in some minor cases, some parts should be reorganized once again.

5.     Uniformity in the listed references must be as per journal guidelines. For further assistance, the authors can use the general guidelines of the journal or its sample template from the relevant official website.

6.     The authors are strongly recommended to round all the numerical values off only up to one point (e.g.: 28.87 → 28.9) throughout the text of the manuscript. Specifically, the related Table should be completely modified.

7.     More updated analytical techniques are required instead of the preliminary phytochemical screening such as preparative HPLC and LC-MS-MS.

8.     Improve the quality of the figures by deleting the subheading inside figures.

Language polishing required 

Author Response

Authors are highly thankful to esteemed reviewers for their excellent comments for further improvement of the article.

We hope the important queries rose by esteemed reviewer # 2 have been answered and revised for their satisfaction. The comments raised by the reviewers have been answered/justified and listed point-by-point below and the corresponding revisions/corrections are highlighted in the revised Manuscript. 

Kindly consider and do the needful.

Thanks and regards,

Dr. Mohamed Rahamathulla

Round 2

Reviewer 2 Report

The authors did all the required corrections